# Immunogenicity and Protection from Receptor-Binding Domains of Toxins as Potential Vaccine Candidates for *Clostridium difficile*

**DOI:** 10.3390/vaccines7040180

**Published:** 2019-11-08

**Authors:** Deyan Luo, Xuechao Liu, Li Xing, Yakun Sun, Jie Huang, Liangyan Zhang, Jiajia Li, Hui Wang

**Affiliations:** Department of Infection Immunity & Defense, State Key Laboratory of Pathogen and Biosecurity, Beijing Institute of Microbiology and Epidemiology, Beijing 100071, China; ldy612@126.com (D.L.); xuechao528@163.com (X.L.); xingli.spring@aliyun.com (L.X.); syk_dyx@hotmail.com (Y.S.); S1006317@163.com (J.H.); polini@live.cn (L.Z.); 13024096923@163.com (J.L.)

**Keywords:** *Clostridium difficile*, receptor-binding domain, toxin, vaccine, protection

## Abstract

The receptor-binding domains (RBDs) located in toxin A and toxin B of *Clostridium difficile* are known to be nontoxic and immunogenic. We need to develop a new type vaccine based on RBDs. In this study, we expressed and purified recombinant proteins (named RBD-TcdA and RBD-TcdB) as vaccine candidates containing the RBDs of toxin A and toxin B, respectively, from the *C. difficile* reference strain VPI10463. The immunogenicity and protection of the vaccine candidates RBD-TcdA, RBD-TcdB, and RBD-TcdA/B was evaluated by ELISA and survival assays. The data indicated that mice immunized with all vaccine candidates displayed potent levels of RBD-specific serum IgG. Following intramuscular immunization of mice with RBD-TcdA and/or RBD-TcdB, these vaccine candidates triggered immune responses that protected mice compared to mice immunized with aluminum hydroxide alone. Taken together, the results of this study reveal that recombinant proteins containing RBDs of *C. difficile* toxins can be used for vaccine development. Additionally, we found that an RBD-TcdA/B vaccine can elicit a stronger humoral immune response and provide better immunoprotection than the univalent vaccines. This RBD vaccine candidate conferred significant protection against disease symptoms and death caused by toxins from a wild-type *C. difficile* strain.

## 1. Introduction 

*Clostridium difficile* or *Clostridiodes difficile* [1] is a gram-positive, spore-forming, anaerobic rod bacterium that barely received attention before it was discovered to be associated with pseudomembranous colitis in 1978 [2,3]. It causes more than 25% of the cases of antibiotic-associated diarrhea [1,2]. The current prognosis of *C. difficile* infection (CDI) is alarming, with a mortality rate between 3% and 15% and a recurrence rate ranging from 12% to 40% [3]. In the United States, CDI is responsible for 500,000 infections [4], approximately 14,000 deaths, and healthcare costs exceeding $3 billion [5] per year.

Most CDIs are caused by a new strain belonging to ribotype 027 (RT027), which spread worldwide in 2003 and resulted in a large number of deaths [6,7]. It has been on the legal surveillance list in many countries, including China, as a pathogen causing infectious disease [8,9]. Besides the use of effective antibiotics and fecal microbiota transplantation (FMT), immunoprophylaxis is generally considered an effective and preferred control strategy [10,11].

Toxin A and toxin B are major virulence determinants of this bacterium. Immunity to these two toxins provides protection by inhibiting the action of the toxins, which can effectively prevent serious illness caused by CDI [6,12]. Toxin A is an enterotoxin and cytotoxin with very high toxicity; toxin B is a potent cytotoxin with a toxicity 100–1000 times as high as that of toxin A [13,14]. In vaccine development, we search for vaccine candidates with high immunogenicity but low toxicity, so we needed to make changes to the toxins for the development of the new vaccine type. Both toxin A and toxin B are single-chain polypeptides consisting of a highly conserved N-terminal catalytic domain that can modify GTPases, a translocation domain, an autoproteolytic domain, and a receptor-binding domain (RBD) [15,16]. The three-dimensional structures of the glucosyltransferase domain and portions of the RBD have been well defined. The only known native receptor of toxin A is the αGal(1,3)βGal(1,4)βGlcgly can sequence, which is not found on human intestinal epithelial cells [16,17]. Human receptors of toxin B have been identified [18,19,20]. Crystal structures of toxin fragments indicate that the toxin A RBD possesses seven carbohydrate binding sites [21], and toxin B is predicted to have four sites [16]. Binding to receptors by RBDs is essential for the toxicity of *C. difficile* toxins A and B; inhibiting the binding can protect a host from illness caused by the toxins. Usually, the RBDs are known to be nontoxic and immunogenic, and RBDs were used in vaccine candidates against, among others, SARS Coronavirus [22] and *C. botulinum* [23]. In our study, we constructed recombinant proteins containing the full-length RBD from toxin A or toxin B (named RBD-TcdA and RBD-TcdB, respectively) of reference strain VPI10463 (ribotype RT087). We immunized mice with the recombinant RBD-TcdA and/or RBD-TcdB to test their immunogenicity and protective effect against toxins extracted from the wild-type strain American Type Culture Collection (ATCC) BAA-1870, which belongs to ribotype 027 (RT027).

## 2. Materials and Methods

### 2.1. Mice, Cells, and Bacteria

All animal studies were conducted in accordance with the Beijing Institute of Microbiology and Epidemiology Animal Care and Use Committee (2012-06-21-02) guidelines. C57BL/6 wild-type mice (6 weeks old, weighing 14–16 g) were obtained from our institute Laboratory Animal Center (Beijing, China). All experimental mice were bred in a specific pathogen-free facility at our institute. The reference strain VPI10463/RT087 and wild-type strain ATCC BAA-1870/RT027 of *C. difficile* were purchased from the American Type Culture Collection (ATCC) center. We sequenced the genes encoding toxin A and toxin B of our wild-type strain. A Vero cell line was kept in our lab.

### 2.2. Protein Expression and Purification

The amino acid sequence corresponding to the RBDs of *C. difficile* RBD-TcdA (strain VPI10463, residue positions 1867–2708) and RBD-TcdB (strain VPI10463, residue positions 1751–2366) were amplified by PCR from the genome of *C. difficile*. These two gene fragments were inserted into a pET-22b vector (Novagen, Darmstadt, Germany) containing a poly-histidine tag at the 3’-end between NdeI and XhoI restriction sites. The resulting pET-22b-RBD-TcdA and pET-22b-RBD-TcdB RBD constructs were transformed into *E. coli* BL21(DE3) (Transgen, Bejing, China) for subsequent protein expression and purification. The recombinant proteins were expressed in successfully transformed bacteria by induction with isopropyl-β-D-thiogalactopyranoside (IPTG) in Luria–Bertani medium and then purified with a Ni^2+^-HiTrap chelating 5 ml prepacked column (GE Healthcare Bio-Sciences Corp, Piscataway, NJ, USA), using imidazole as the elution reagent. The lysates of transformed cells and the purified proteins were identified by sodium dodecylsulfate-polyacrylamide gel electrophoresis (SDS-PAGE) and Western blot.

### 2.3. SDS-PAGE and Western Blot

All purification steps were analyzed by 8% SDS-PAGE. The RBD-TcdA and RBD-TcdB were separated. Samples separated in the gel were electrotransferred to a PVDF membrane (GE). This step was followed by the antigen–antibody reactions. Rabbit anti-*C. difficile* hyper-immune sera were used as the detecting antibody, and horseradish peroxidase-conjugated goat anti-rabbit immunoglobulin G (IgG) served as the secondary antibody. Following the addition of the substrate diaminobenzidine, the specific protein bands were revealed.

### 2.4. Crude Toxins Preparation and Toxicity Analysis

Two *C. difficile* strains were grown in brain heart infusion (BHI, Becton, Dickinson Company) at 37 °C for 72 h. Cells were removed by centrifugation. The culture supernatant was precipitated by 60% saturated (NH_4_)_2_SO_4_ overnight at 4 °C and then centrifuged at 5000 rpm for 30 min. The precipitate was dissolved in 10 mM Tris-HCl (pH 7.5) and dialyzed in 10 mM PBS (pH 7.2) at 4 °C for 24 h. Toxins were filtration-sterilized through 0.22 μm (pore size) membranes and stored at −20 °C. The samples were analyzed by SDS-PAGE and Western blot. Toxin A and toxin B were purified using DEAE SepharoseFF, and their molecular weights were measured using the native-PAGE method. Toxicity was determined by using Vero cells and mice. In brief, Vero cells were seeded into 96-well plates at a density of 1.0 × 10^5^ cells per well. Incubation of Vero cells with crude toxins resulted in a loss of cell adherence and a change in cell morphology, which was observed under microscope to calculate the dose that causes 50% of the Vero cells to round (ED_50_).

For the in vivo experiments, we treated mice (10 mice/group, 10 groups) with serial tenfold diluents of 100 μg crude toxins to monitor the survival rate. The minimal dose that caused complete lethality (MLD) in mice was calculated with the Reed–Muench method.

### 2.5. Mouse Immunization and Toxin Challenge

C57BL/6 mice were immunized intramuscularly with different doses (1 µg and 10 µg) of RBD-TcdA, RBD-TcdB, and RBD-TcA/TcdB (50% of each) on day 0 and day 14 with aluminum hydroxide as adjuvant (75 mice for the harvest experiments and 30 mice for the challenge assays). Controls were vaccinated with aluminum hydroxide in PBS. Serum samples for serological analysis were collected on days 7, 21, 49, 80, and 110. Mice were challenged with crude toxins of ATCC BAA-1870 on day 28 by intraperitoneal injection of the crude toxins. The doses ranged from 1 MLD to 3 MLD. Mice were monitored daily. Unresponsive or recumbent animals were considered moribund and euthanized. Moribund mice were dissected for pathologic analysis of the small intestine.

### 2.6. Analysis of Antigen-Specific IgG and Neutralizing Antibody in the Sera of Immunized Animals

The presence of serum IgG and of the subtypes IgG1, IgG2a, IgG2b, and IgG3 specific to vaccine candidates was determined by indirect ELISA. Briefly, the crude toxins of *C. difficile* (strain VPI10463) were diluted to 3 µg/ml in carbonate buffer (pH 9.6) and used to coat the wells. After overnight incubation at 4 °C, the plates were washed, blocked, and then incubated with serially diluted sera for 2 h at room temperature. Following another wash, IgG or isotype-specific rabbit anti-mouse horseradish peroxidase conjugates were added (50 µL/well) at the appropriate dilutions. After 30 min of incubation at room temperature, the plates were washed, 100 µL of substrate solution was added to each well, and the reaction was stopped by the addition of 50 µL of 2 M sulfuric acid to each well. The absorbance of the developed color was measured at 450 nm (A450). The cut-off value for the assay was calculated as the mean specific OD plus standard deviation (SD) for 10 serum samples, assayed at a dilution of 1:40 from nonimmunized mice. The titer of each serum was calculated as the reciprocal of the highest serum dilution yielding a specific OD higher than the cut-off value. All assays were performed in triplicate and repeated three times.

The neutralizing antibody detection was performed by MTT assay. Briefly, the heat-inactivated serum from immunized mice was serially diluted with DMEM containing 10% fetal calf serum, mixed with an equal volume of crude toxins, i.e., 0.28 ng (50 μL) toxin A, or 0.36 ng (50 μL) toxin B, and incubated at 37 °C for 1 h. The mixture was added to the 96-well plates containing Vero cells and incubated in 5% CO_2_ at 37 °C for 24 h. The result was detected by MTT staining of toxin-treated Vero cells after discarding the non-adherent cells. The plates were read on a microtiter plate reader at a wavelength of 490 nm.

### 2.7. Histology

Intestinal tissues were fixed in 10% neutral buffered formalin, embedded in paraffin, sectioned, and stained with hematoxylin and eosin. The pathological foci (i.e., areas with large numbers of inflammatory cell infiltration accompanied by evidence of edema of submucosa) in each section were evaluated. Representative photomicrographs were taken at × 200 magnification.

### 2.8. Statistical Analysis

Statistical analyses were performed using the program Prism5.0 (GraphPad Software, Inc., LaJolla, CA, USA). Values are expressed as mean ± SD. Data were analyzed by unpaired Student’s *t*-test (normal distribution) or one-way ANOVA followed by Dunnett’s multiple comparison test. Survival data were analyzed by log-rank tests. Values of *p* < 0.05 are considered to be statistically significant.

## 3. Results

### 3.1. Expression and Purification of Proteins RBD-TcdA and RBD-TcdB

To evaluate the relative roles of two RBD vaccines in inducing an immune response and protective immunity against the toxins, we expressed and purified the two proteins RBD-TcdA and RBD-TcdB. Fragments encoding the C-terminal RBD of toxin A or toxin B were generated to produce recombinant proteins, which were, respectively, inserted into a pET-22b vector containing a 6×His-tag at the3’-end. The successful insertion of the gene fragments into the plasmids was verified with restriction digestions and PCR (Figure 1A,C). The recombinant proteins RBD-TcdA and RBD-TcdB were purified by Ni-affinity chromatography. The purification process yielded products of a purity over 90%. When the purified proteins were visualized by Western blot, a 96-kDa protein and a 72-kDa protein appeared, as shown in Figure 1B,D. These molecular masses correspond to those of RBD-TcdA and RBD-TcdB, respectively.

### 3.2. Crude Toxins Extracted from ATCC BAA-1870

We chose to challenge immunized mice, in our study, with crude toxins extracted from bacteria, because mice are not sensitive to CDI. To obtain crude toxin A and toxin B of ATCC BAA-1870, we have first precipitated the culture supernatant with 60% (NH_4_)_2_SO_4_. We obtained 86.5mg/L crude toxins. Toxin A and toxin B were purified using DEAE Sepharose FF (Figure 2A), and their molecular masses were measured using native-PAGE. The purified proteins were stored at −70 °C or –20 °C as mentioned in the method section, until we used it for the ELISA experiment. As shown in Figure 2B, both toxins showed the correct molecular masses and the identities were confirmed by ELISA (Toxin A OD450 = 0.910, Toxin B OD450 = 0.660, mean OD450 = 0.116). We challenged mice and cells using crude toxins to test their toxicity. The MLD in mice is 540 ng for ATCC BAA-1870, and the ED_50_ for Vero cells is 90 pg (Figure 2C).

### 3.3. Immunogenicity of RBD-TcdA and RBD-TcdB Recombinant Proteins

In humans, strong humoral toxin-specific immune responses elicited by CDI are associated with recovery and lack of disease recurrence, whereas insufficient humoral responses are associated with recurrent CDI. Therefore, in our study, we tested the humoral immune responses through evaluating toxin-specific antibody titers. To evaluate humoral toxin-specific immune responses elicited by RBD-TcdA and RBD-TcdB, mice were vaccinated twice intramuscularly with 1 μg RBD-TcdA, RBD-TcdB, or RBD-TcdA/B. On the indicated days, serum samples were collected and evaluated by ELISA. The data indicate that all immunized mice displayed potent levels of RBD-specific serum IgG (Figure 3A,B). As expected, animals immunized with PBS displayed no antigen-specific responses (data not shown). The anti-RBD-TcdA and anti-RBD-TcdB antibody titers can be detected on day 7, and then increased quickly and maintained a high-level titer for a few days (Figure 3A,B). When we increased the immunization dose to 10 µg of RBD-TcdA, RBD-TcdB, or RBD-TcdA/B, we did not measure higher titers of IgG in comparison with groups immunized with lower doses (Figure 3C,D). The presence of antibody subtypes (IgG1, IgG2a, IgG2b, and IgG3) specific to vaccine candidates was determined by indirect ELISA. We found significant increases in absorbance for antigen-specific IgG1, IgG2a, and IgG2b (Figure 3E,F), but not IgG3, in the sera of immunized mice in comparison with controls. Next, we evaluated the neutralizing antibody levels. Sera from immunized mice neutralized a certain dose of crude toxins in our in vitro neutralization assay. The titers of neutralizing antibody induced by the RBD-TcdA/B vaccine were much higher than those of the two univalent vaccines (Table 1).

### 3.4. Vaccine Candidates Improve Survival Rate after Inoculation of Lethal Dose of Crude Toxins

Although the vaccine approach holds great potential for CDI prevention, immunoprotection against CDI had not yet been achieved. Since we used the toxins of wild-type ATCC BAA-1870 strain of *C. difficile* to challenge mice in this study, we needed to verify the similarities between the toxins of ATCC BAA-1870 and other hyper-virulent strains R20291 (Toxin A: Gene ID: 8470358; Toxin B: Gene ID: 8470357) and CD196 (Toxin A: Gene ID: 8466712; Toxin B: Gene ID: 8466711). The results show that genes of both toxins in the ATCC BAA-1870 strain are identical to those in the other two RT 027 strains, as published in GenBank. However, the RBDs amino acid sequences of toxins A and B of the ATCC BAA-1870 strain show 96.2% and 88.4% similarity with those of the reference strain VPI10463, respectively (Appendix A). To study the protective effect of the recombinant proteins, C57BL/6 mice were immunized with 1 µg RBD-TcdA, RBD-TcdB, or RBD-TcdA/B and, after two immunizations, inoculated with crude toxins of ATCC BAA-1870. The results show that RBD-TcdA, RBD-TcdB, and RBD-TcdA/B can protect mice against 1 MLD of toxin or prolong their life (Figure 4A). When we increased the immunization dose to 10 µg, these three vaccines still protected mice against 1 MLD of crude toxins or prolonged their life (Figure 4B). There is no big difference between the effects of the two immunization doses when mice were challenged with 1 MLD. When mice were challenged with a dose of 2 MLD crude toxins, 80% of the mice immunized with RBD-TcdA/B survived. Of the mice immunized with RBD-TcdA, 50% survived, and in the RBD-TcdB-vaccinated group and the control group, there were no survivors. When mice were challenged with a dose of 3 MLD crude toxins, 60% of the mice immunized with 10 µg RBD-TcdA/B survived; only the group receiving 10 μg RBD-TcdA/B had survivors. (Table 2) 

### 3.5. Vaccine Candidates Reduce Intestinal Pathology After Inoculation of Lethal Dose of Crude Toxins

Histological evaluations revealed an intestinal pathology typical of toxin, with the mice that had received a high dose of toxins showing many inflammatory foci with various degrees of cellular damage and evidence of edema of submucosa (Figure 5). The main differences between the experimental groups were the following. In the nonimmunized mice, the cellular infiltrates tended to be of medium to large size and were frequently associated with large areas of necrosis and edema of submucosa (Figure 5B). The mice immunized with 1 µg RBD-TcdA/B (Figure 5C) or 10 µg RBD-TcdA/B (Figure 5D) and challenged with 1 MLD crude toxins displayed sporadic, small sites of inflammation. The mice immunized with 1 µg RBD-TcdA/B (Figure 5E) or 10 µg RBD-TcdA/B (Figure 5F) and challenged with 2 MLD crude toxins displayed sporadic sites of inflammation, which were a little bigger than those of mice challenged with 1 MLD crude toxins. Quantitative scoring by an expert revealed that a significantly greater percentage of the lesions in the unimmunized mice showed evidence of necrosis and edema of submucosa (*p* < 0.05).

## 4. Discussion

In recent years, there has been an increase in the occurrence of CDI, which is related to certain risk factors, especially the use of antibiotics. Usually, the treatment of CDI is costly, and the recurrence rate is very high [5,7]. In recent years, FMT has been a promising method for CDI treatment [24]. There are limited data on FMT for treatment of primary CDI, but FMT appears safe and effective for recurrent CDI [24]. Disease prevention is still an important approach that deserves exploration [25,26]. As mentioned before, toxin A and toxin B are major virulence determinants of *C. difficile*. Binding to receptors by RBDs is essential for the toxicity, and blocking the binding sites can protect hosts from illness caused by toxins [2,14]. Immunization with recombinant RBDs represents a promising method in vaccine development [22,23]. A number of studies have demonstrated that antibodies induced by RBD-based vaccines can efficiently block the toxicity of bacterial or viral replication in tissue [27]. Several *C. difficile* vaccine candidates exist worldwide [27,28,29]. One vaccine for *C. difficile* was developed by Sanofi Pasteur [30]. They developed a bivalent candidate vaccine, a formalin-inactivated, highly purified preparation of toxins. Although a significant immune response is evoked, it remains quite difficult to develop this kind of vaccine, especially because it is difficult to obtain a highly pure natural toxin. Novavax developed a chimeric toxin receptor-binding domains vaccine, which induced protection against CDI [31]. The linker in chimeric vaccine may alter the structure of protein. Sometimes, simple design is more useful for vaccine development. An optimized, synthetic DNA vaccine encoding the toxin A and toxin B receptor-binding domains of *C. difficile* induced protective antibody [31], but DNA vaccine is not the best optimum prophylaxis for an extracellular bacterium infection. One nontoxigenic *C. difficile* strain expressing mTcd138 provided mice full protection against infection with a hypervirulent *C. difficile* strain, UK6 (ribotype 027) [32]. It is a promising oral vaccine candidate against CD. As mentioned in this paper, they still need a long time to evaluate whether NTCD/NTCD_mTcd138 survived in animals after antibiotic treatment. Another strategy is developing a novel chimeric toxin vaccine, which retains major neutralizing epitopes from both toxins and confers complete protection [32]. The subunit vaccine will become the most promising measure.

Therefore, we designed recombinant RBDs of reference strain VPI10463 as vaccine candidates in this study. We immunized mice with recombinant RBDs and challenged them with extracted crude toxins from a hyper-virulent strain. Several variations of the hyper-virulent strain RT 027 exist, which cause high infection rate and drug resistance. Consequently, the morbidity and mortality rates of this strain are higher than those of other strains [8,33]. Next, we needed to evaluate the protection against toxins of the *C. difficile* ATCC BAA-1870 strain conferred by vaccines targeting the RBDs of toxin A and/or toxin B from the *C. difficile* reference strain VPI10463.

Our results confirmed that both RBD-TcdA and RBD-TcdB are highly immunogenic for mice [34]. To investigate which subtype is predominant in mice immunized with vaccine candidates, the IgG subtypes were assayed by ELISA. We found a significant increase in absorbance for antigen-specific IgG1, IgG2a, and IgG2b but not IgG3. The ratio of IgG2a/IgG1 in all vaccine groups suggested that our vaccines elicited a Th2-type immune response. The principal role of the toxin-neutralizing circulating antibody in immunity against *C. difficile* has also been clearly demonstrated in animal models [27,35,36]. In this study, we identified and characterized efficacious toxin RBD-neutralizing antibodies against crude toxins. Sera from immunized mice neutralized a certain dose of crude toxins in an in vitro neutralization assay. The neutralizing antibody titer induced by the RBD-TcdA/B vaccine was much higher than those induced by the two univalent vaccines.

We could not set up the stable challenge animal model with spores of bacterium stain ATCC BAA-1870, whereas we still stuck to the use of crude toxins of this strain for challenge only because it has repeatedly appeared in some Chinese hospitals over the years. Delivery of crude toxins intraperitoneally is lethal in mice. However, systemic toxin is not indicative of a normal infection scenario [37]. Entry of the toxin into circulation is thought to be a possible cause [38]. Therefore, challenging immunized mice with intraperitoneal toxin represents a stringent method for this research. However, we will try to make spores of bacterium to establish the animal CDI model, which can develop a wide range of disease symptoms in the near future. In order to evaluate the protection conferred by the vaccine candidate, the immunized mice were challenged with natural crude toxins extracted from the RT027 strain ATCC BAA-1870 [36]. Among the three vaccine candidates, the RBD-TcdA/B vaccine elicited the strongest humoral immune response and provided the greatest protection against toxins of *C. difficile*. The most important reason for this observation must be that the RBD-TcdA/B vaccine can induce the production of more types and higher levels of RBD-specific antibodies, which can block two toxin RBDs to inhibit the action of the toxins. Moreover, toxins can be used as adjuvant in several research models [39,40]. In our study, we found that the anti-toxin A, but not anti-toxin B, neutralizing antibody titer induced by the RBD-TcdA/B vaccine was much higher than that induced by the single one. Toxin B is a potent cytotoxin with toxicity 100–1000 times as high as that of toxin A. Thus, we presume that RBD-TcdB functions as an adjuvant here. In this case, the RBD-TcdA/B vaccine can activate a broader spectrum of B-cell clones than the univalent vaccine, causing a difference in the protection level they offer against toxins. The protective data we got from the animal experiments will aid us further in the development of a safe and effective vaccine for human beings. C57BL/6 mice were also immunized with different doses to evaluate the relationship between the immunizing dose and the conferred protection. The results show that there is no statistically significant difference between the immunization effects of the two doses after mice were challenged with 1 MLD crude toxin. We need more work to address it and find the mechanism.

## 5. Conclusions

In this study, it is proved for the first time that a vaccine based on RBDs can induce a significant humoral immune response to *C. difficile*. Our data decisively demonstrate that even a small dose of the bivalent vaccine candidate conferred remarkable protection against toxins of *C. difficile* on the host, indicating that the RBDs can be used as a relatively efficacious target for immunization.

## Figures and Tables

**Figure 1 vaccines-07-00180-f001:**
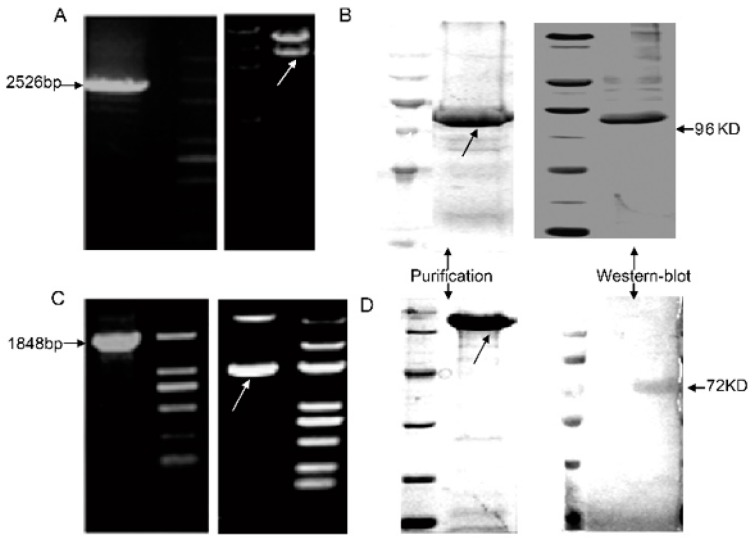
Construction and expression of toxin receptor-binding domains (RBDs) of *C. difficile* reference strain VPI10463. (**A**) PCR product and enzymatic digestion analysis of the pET-22b-RBD-TcdA plasmid. Lane 1, PCR product. Lane 4, pET-22b-RBD-TcdA plasmid digested by restriction enzymes. (**B**) The purification and Western blot identification of the recombinant protein RBD-TcdA are shown (molecular mass is indicated on the right). (**C**) PCR product and enzymatic digestion analysis of the pET-22b-RBD-TcdB plasmid. Lane 1, PCR product. Lane 4, pET-22b-RBD-TcdB plasmid digested by restriction enzymes. (**D**) The purification and Western blot identification of the recombinant protein RBD-TcdB are shown (molecular mass is indicated on the right).

**Figure 2 vaccines-07-00180-f002:**
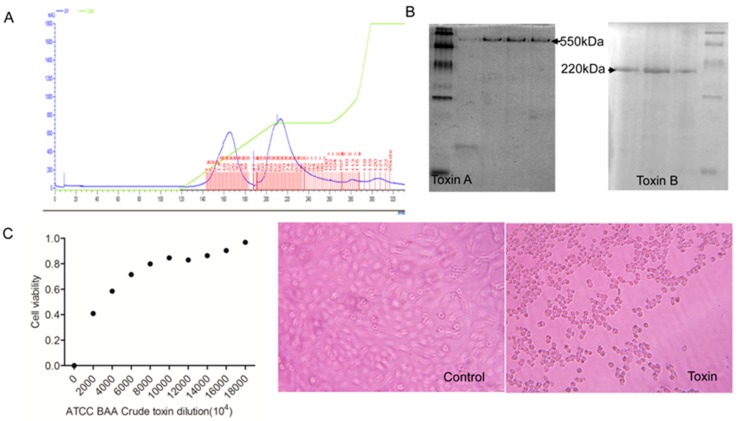
Activity analysis of crude toxins of wild-type strain *C. difficile* ATCC BAA-1870. (**A**) Purification of crude toxins of *C. difficile* ATCC BAA-1870. The first peak is toxin A, the second peak is toxin B. (**B**) Native-PAGE of purified toxins (predicted TcdA and TcdB molecular weights are 550 KD and 220 KD). (**C**) Toxicity of crude toxins was determined by Vero cell viability analysis.

**Figure 3 vaccines-07-00180-f003:**
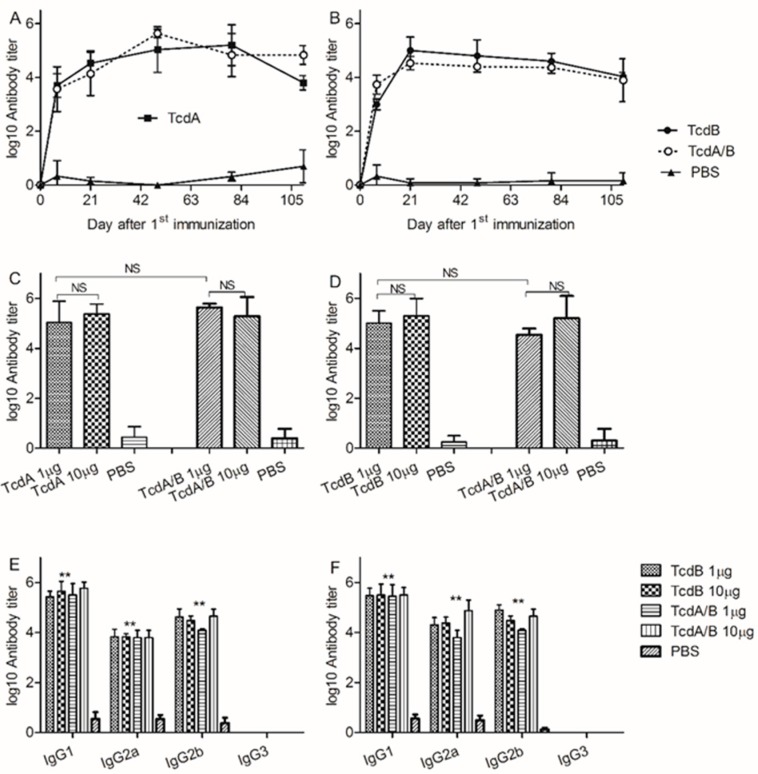
The immunogenicity of vaccine candidates. (**A**) Anti-RBD-TcdA antibody titers at different time points (1 µg group). (**B**) Anti-RBD-TcdB antibody titers at different time points (1 µg group). (**C**) Anti-RBD-TcdA antibody titers on day 49. (**D**) Anti-RBD-TcdB antibody titers on day 21. (**E**) Antibody subtypes of mice immunized with RBD-TcdA. (**F**) Antibody subtypes of mice immunized with RBD-TcdB. Statistics were performed using the Student’s *t*-test or ANOVA followed by Dunnett’s multiple comparison test (** *p* < 0.01; NS, no significance). Similar results were observed in two independent experiments.

**Figure 4 vaccines-07-00180-f004:**
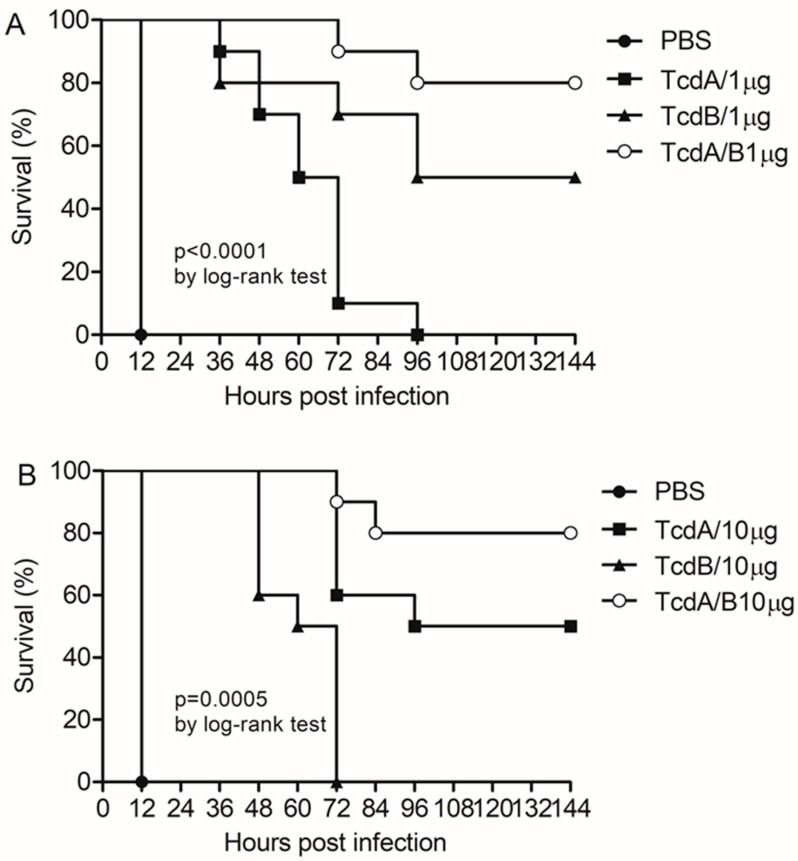
Vaccine candidates protected mice against lethal dose of toxins. (**A**) Survival rates of wild-type C57BL/6 mice immunized with 1 µg recombinant vaccines and challenged with 1 MLD crude toxins. In comparison with mice immunized with aluminum hydroxide, all vaccine candidates increased the survival rate (*p* < 0.001; *n* = 8–10 mice/group). (**B**) Survival rate for wild-type C57BL/6 mice immunized with 10 µg recombinant vaccines and challenged with 1 MLD crude toxins. In comparison with mice immunized with aluminum hydroxide, all vaccine candidates increased the survival rate (*p* < 0.001; *n* = 8–10 mice/group). Similar results were observed in two independent experiments. Survival data were analyzed by log-rank tests.

**Figure 5 vaccines-07-00180-f005:**
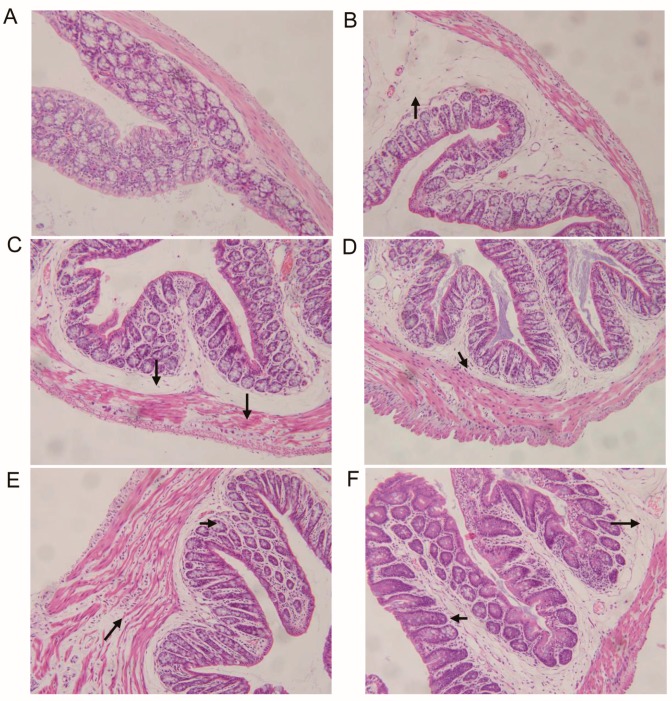
Vaccine immunization impacts intestinal histopathology after challenge. In comparison with healthy C57BL/6 mice (**A**), nonimmunized C57BL/6 mice challenged with toxins (**B**) displayed sporadic large sites of inflammation. The mice immunized with 1 µg RBD-TcdA/B (**C**) or 10 µg RBD-TcdA/B (**D**) and challenged with 1 MLD crude toxins displayed sporadic small sites of inflammation. The mice immunized with 1 µg RBD-TcdA/B (**E**) or 10 µg RBD-TcdA/B (**F**) and challenged with 2 MLD crude toxins displayed bigger sites of inflammation than mice challenged with 1 MLD. Similar results were observed in two independent experiments.

**Table 1 vaccines-07-00180-t001:** Vaccine candidates induced high titers of neutralizing antibody.

Group		Neutralizing Antibody Titer
		ED_50_ (AVG)
1 µg	RBD-TcdA	1:529 (anti-toxin A)
	RBD-TcdB	1:1223 (anti-toxin B)
	RBD-TcdA/B	1:1621 * (anti-toxin A) 1:1310 (anti-toxin B)
10 µg	RBD-TcdA	1:1516 (anti-toxin A)
	RBD-TcdB	1:933 (anti-toxin B)
	RBD-TcdA/B	1:6410 ** (anti-toxin A) 1:1010 (anti-toxin B)

* *p* < 0.05, vs. 1 μg RBD-TcdA immunized group; ** *p* < 0.01, vs. 10 μg RBD-TcdB immunized group; ED_50_(AVG) = 90 pg.

**Table 2 vaccines-07-00180-t002:** Protection conferred by different vaccine candidates against toxins.

Immunization Dose	Group	Survival (%)		
		1MLD	2MLD	3MLD
1 µg	RBD-TcdA	100% (10/10)	0% (0/10)	0% (0/10)
	RBD-TcdB	50% (5/10)	50% (5/10)	0% (0/10)
	RBD-TcdA/B	80% (8/10)	80% (8/10)	0% (0/10)
10 µg	RBD-TcdA	50% (5/10)	0% (0/10)	0% (0/10)
	RBD-TcdB	100% (10/10)	0% (0/10)	0% (0/10)
	RBD-TcdA/B	80% (8/10)	80% (8/10)	60% (6/10)

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
