# Peer review of "Immunogenicity and Protection from Receptor-Binding Domains of Toxins as Potential Vaccine Candidates for *Clostridium difficile"

_vaccines, 2019, doi:10.3390/vaccines7040180_

Round 1
Reviewer 1 Report
Major comment:
The paper from Luo et al. describes the use of recombinant proteins containing RBDs of C. difficile toxin A and B as vaccine candidates.
This paper reports promising data; however it should more highlight the interest of this study compared to previous ones.
Specific comments:
Abstract:
Line 26: “we expressed and purified recombinant fusion proteins”: vaccine candidates used in this study do not correspond to fusion proteins but only recombinant proteins: please correct.
Line 35 to 37: This study is not the first revealing the interest of recombinant proteins containing RBDs of C. difficile toxins as vaccine candidate. Please moderate this aspect. Other publications mentioned this strategy with interesting results, even if it used fusion proteins (J.H. Tian, S.R. Fuhrmann, S. Kluepfel-Stahl, R.J. Carman, L. Ellingsworth, D.C. Flyer. A novel fusion protein containing the receptor binding domains of C. difficile toxin A and toxin B elicits protective immunity against lethal toxin and spore challenge in preclinical efficacy models. Vaccine 30 (2012) 4249-58)
Introduction:
Line 58: the reference 14 is related to C. difficile binary toxin (CDT) which is not mentioned above, please change this inappropriate citation or mentioned this third C. difficile toxin before.
Line 68: the authors mentioned that: “Human receptors of toxin B have not yet been identified but its main binding target is thought to be a disaccharide moiety”. This information is not updated (Yuan et al, 2015, Cell ERes 25:157-168; Lafrance et al, 2015, Proc Natl Acad Sci USA 112:7073-7078 and Tao et al, 2016, Nature 538:350-355)
Line 78: the strain ATCC 43255 is also known (and published) as strain VPI 10463.
Methods
2.4: The in vivo experiments of mice treated with different doses of crude toxins must be more detailed: which dose of crude toxins? How many mice treated?
2.5: How many animals have been treated?
Results:
The titer 3.1 mentioned fusion proteins but according to method section, the proteins correspond to recombinant proteins but not fusion proteins: please clarify that point.
Figure 1B: images corresponding to protein expression gel do not seem necessary; purification and western-blot figure are enough. Wester-blot image quality is not sufficient.
Line 205: “Crude toxin” please replace by “crude toxins”
Line 207: please replace natural by native or crude
Line 211 please replace “The purified protein was stored at 70°C” by “purified proteins were stored at: -70°C or -20°C as mentioned in method section?”
Figure 2A is not mentioned in the text.
Figure 2, line 218: the molecular weight usually mentioned for C. difficile toxins A and B is 308 and 270 kDa, respectively (PNAS | July 27, 2010 | vol. 107 | no. 30 | 13467–13472), please explain the difference with the cited: « Predicted TcdA and TcdB molecular weights are 550KD and 220KD”
Line 218: Toxicity of crude “toxins”
Line 220: please remove “fusion”
Figure 3A: there is a problem with the legend
Figure 3C and D: could the authors explain why doses of 1mg and 10 mg are mentioned here and not in method section?
Figure 3E: there is a problem with the legend
Figure 3E and F: are statistic tests been performed? Is yes, what are the results?
Table 1: the author mentioned a statistical difference for RBD-TcdA/B but what is the reference group? RBD-TcdA or RBD-TcdB? And ED50 (AVG) must be define
Line 259: the sentence is not clear.
Line 261: Could the authors explain why the dose used for the protection assay is 0.1µg which is not the same as the dose used in the immunological assays?
According to figure 4 A and B, groups immunized with TcdA/1µg and TcdB/10µg are not protected, on the contrary to what is written in the text (line 265 to 268).
Line 275: this sentence should be clarified.
Discussion:
Line 329: “The most famous vaccine for C. difficile is the one developed by Sanofi Pasteur » the reference cited (27) is not the most appropriate of the Sanofi Pasteur C. difficile vaccine development ref 27 is about the fully human monoclonal antibodies. The most developed vaccine is a toxoid vaccine. They started a phase III clinical development (now stopped).
Line 365: Authors mentioned that: “Our results showed that both RBD TcdA and RBD TcdB are highly immunogenic for mice.” Other studies have also studied on the immunogenicity of tcdA and TcdB fragment. So authors should writte: “our results confirmed that…..(R. Leuzzi, J. Spencer, A. Buckley, C. Brettoni, M. Martinelli, L. Tulli, et al. Protective efficacy induced by recombinant Clostridium difficile toxin fragments. Infect Immun 81 (2013) 2851-60.)
Could the authors mention and discuss the results presented in this paper and compare to that previously published? (H. Wang, X. Sun, Y. Zhang, S. Li, K. Chen, L. Shi, et al. A chimeric toxin vaccine protects against primary and recurrent Clostridium difficile infection. Infect Immun 80 (2012) 2678-88.)
Author Response
Dear reviewer:
We thank your good comments and constructive criticism and believe that we have addressed all of your concerns. We hope our findings have significance for those studying about receptor-binding domains of Clostridium difficile toxins can be used for vaccine development against this infection.
Response to Reviewer 1 Comments
Abstract:
1. You expressed your concern about “Line 26: “we expressed and purified recombinant fusion proteins”: vaccine candidates used in this study do not correspond to fusion proteins but only recombinant proteins: please correct.”
Response: We deleted this word, please see line 26.
2. You expressed your concern about “Line 35 to 37: This study is not the first revealing the interest of recombinant proteins containing RBDs of C. difficile toxins as vaccine candidate. Please moderate this aspect. Other publications mentioned this strategy with interesting results, even if it used fusion proteins (J.H. Tian, S.R. Fuhrmann, S. Kluepfel-Stahl, R.J. Carman, L. Ellingsworth, D.C. Flyer. A novel fusion protein containing the receptor binding domains of C. difficile toxin A and toxin B elicits protective immunity against lethal toxin and spore challenge in preclinical efficacy models. Vaccine 30 (2012) 4249-58)”
Response: We moderated this aspect, please see line 35.
Introduction:
1. You expressed your concern about “Line 58: the reference 14 is related to C. difficile binary toxin (CDT) which is not mentioned above, please change this inappropriate citation or mentioned this third C. difficile toxin before.”
Response: We changed this inappropriate citation, please see line 62.
2. You expressed your concern about “Line 68: the authors mentioned that: “Human receptors of toxin B have not yet been identified but its main binding target is thought to be a disaccharide moiety”. This information is not updated (Yuan et al, 2015, Cell ERes 25:157-168; Lafrance et al, 2015, Proc Natl Acad Sci USA 112:7073-7078 and Tao et al, 2016, Nature 538:350-355)”
Response: That is really big help, I corrected our description, please see line 83.
3. You expressed your concern about “Line 78: the strain ATCC 43255 is also known (and published) as strain VPI 10463.”
Response: We changed to VPI10463 in our manuscript.
Methods:
1. You expressed your concern about “2.4: The in vivo experiments of mice treated with different doses of crude toxins must be more detailed: which dose of crude toxins? How many mice treated?”
Response: We added more detailed about dose and the number of mice in the method, please see line 159.
2. You expressed your concern about “2.5: How many animals have been treated?”
Response: We revised this section, please see line 167.
Results:
1. You expressed your concern about “The titer 3.1 mentioned fusion proteins but according to method section, the proteins correspond to recombinant proteins but not fusion proteins: please clarify that point.”
Response: Yes, we clarified in whole manuscript.
2. You expressed your concern about “Figure 1B: images corresponding to protein expression gel do not seem necessary; purification and western-blot figure are enough. Wester-blot image quality is not sufficient.”
Response: We totally agree with you, we deleted protein expression gel images, and improved the WB image quality, please see Fig 1.
3. You expressed your concern about “Line 205: “Crude toxin” please replace by “crude toxins””
Response: We replaced “Crude toxin” by “crude toxin”. Please see line 241.
4. You expressed your concern about “Line 207: please replace natural by native or crude”
Response: We replaced “natural ”by “crude”.Please see line 243.
5. You expressed your concern about “Line 211 please replace “The purified protein was stored at 70°C” by “purified proteins were stored at: -70°C or -20°C as mentioned in method section?””
Response: We replaced “The purified protein was stored at 70°C” by “purified proteins were stored at: -70°C or -20°C as mentioned in method section. Thank you for your nice corrections. Please see line 247.
6. You expressed your concern about “Figure 2A is not mentioned in the text.“
Response: We added Figure 2A in the text, please see line 246.
7. You expressed your concern about “Figure 2, line 218: the molecular weight usually mentioned for C. difficile toxins A and B is 308 and 270 kDa, respectively (PNAS | July 27, 2010 | vol. 107 | no. 30 | 13467–13472), please explain the difference with the cited: ? Predicted TcdA and TcdB molecular weights are 550KD and 220KD” ”
Response: The molecular weight of crude toxins are 308kDa and 270kDa, but when we purified the crude toxins, the molecular weights would changed in all publications. For example, the molecular weight of Toxin B, please see the reference “Purification and Characterization of Toxin B from Clostridium difficile”. The structure of protein changed.
8. You expressed your concern about “Line 218: Toxicity of crude “toxins” ”
Response: We added a “s”.
9. You expressed your concern about “Line 220: please remove “fusion” ”
Response: We removed this word in all manuscipt.
10. You expressed your concern about “Figure 3A: there is a problem with the legend”
Response: We revised this legend, please see line 291.
11. You expressed your concern about “Figure 3C and D: could the authors explain why doses of 1mg and 10 mg are mentioned here and not in method section? ”
Response: Sorry, this is a mistake, should be 1mg and 10mg. We corrected, please see Figure 3C and D.
12. You expressed your concern about “Figure 3E: there is a problem with the legend”
Response: We revised this legend, please see line 294.
13. You expressed your concern about “Figure 3E and F: are statistic tests been performed? Is yes, what are the results?”
Response: Yes, we did. Please see Figure 3E and F.
14. You expressed your concern about “Table 1: the author mentioned a statistical difference for RBD-TcdA/B but what is the reference group? RBD-TcdA or RBD-TcdB? And ED50 (AVG) must be define”
Response: We added under the table 1.ED50(AVG)=90pg. Please see line 308.
15. You expressed your concern about “Line 259: the sentence is not clear. ”
Response: We corrected this sentence, please see line 313.
16. You expressed your concern about “Line 261: Could the authors explain why the dose used for the protection assay is 0.1μg which is not the same as the dose used in the immunological assays? ”
Response: Yes, I totally agree with your suggestion, it is no use to mention in protection assay even though we did it, we deleted it. Please see line 326.
17. You expressed your concern about “According to figure 4 A and B, groups immunized with TcdA/1μg and TcdB/10μg are not protected, on the contrary to what is written in the text (line 265 to 268). ”
Response: Sorry for the unclearly description. Mice immunized with TcdA/1μg and TcdB/10μg did prolong their life in comparison with mice immunized aluminum hydroxide, but no survivors at last. We revised our description, and added “prolong their life”. Please see 328,330.
18. You expressed your concern about “Line 275: this sentence should be clarified. ”
Response: We clarified this sentence, please see line 336.
Discussion:
1. You expressed your concern about “Line 329: “The most famous vaccine for C. difficile is the one developed by Sanofi Pasteur ? the reference cited (27) is not the most appropriate of the Sanofi Pasteur C. difficile vaccine development ref 27 is about the fully human monoclonal antibodies. The most developed vaccine is a toxoid vaccine. They started a phase III clinical development (now stopped). ”
Response: Sorry, this is really a mistake, we corrected, and please see line 402.
2. You expressed your concern about “Line 365: Authors mentioned that: “Our results showed that both RBD TcdA and RBD TcdB are highly immunogenic for mice.” Other studies have also studied on the immunogenicity of tcdA and TcdB fragment. So authors should writte: “our results confirmed that…..(R. Leuzzi, J. Spencer, A. Buckley, C. Brettoni, M. Martinelli, L. Tulli, et al. Protective efficacy induced by recombinant Clostridium difficile toxin fragments. Infect Immun 81 (2013) 2851-60.) ”
Response: We corrected our description, and cited this reference, please see line 439.
3. You expressed your concern about “Could the authors mention and discuss the results presented in this paper and compare to that previously published? (H. Wang, X. Sun, Y. Zhang, S. Li, K. Chen, L. Shi, et al. A chimeric toxin vaccine protects against primary and recurrent Clostridium difficile infection. Infect Immun 80 (2012) 2678-88.) ”
Response: We did not, we mentioned and discussed this results of this paper, and cited this reference, please see line 426. Thank you for your nice suggestions.
Reviewer 2 Report
Review of the paper vaccines 594412 by Luo et al.
In this paper, the authors vaccinate mice with the C-terminal domain of C. difficile Toxin and Toxin B or a combination thereof. They measure the antibody responses in the mice by ELISA. In addition, they tested whether the induced antibodies were able to neutralize toxicity of Toxin A and B in a cell based assay and they determined the type of IgG molecules were induced. Finally ,they “challenged” the immunized mice by intraperitoneal injection with purified toxins and see protection.
Several important issues appear when reading this paper.
1) The animal model which the authors use does not mimic a proper C. difficile infection. Although the authors claim that “mice are not sensitive to CDI”, excellent mouse models have been described by for example the group of Vince Young, in which even recurrent CDI can be mimicked (see for instance Seekatz et al., Infect Immun. 2015 Oct;83(10):3838-46). The problem with the model is that purified toxins are injected into the peritoneum of the mice. In a normal CDI, the toxins are not present in the peritoneum and therefore, the model is not representative for CDI. Also, the target of the C. difficile toxins is the colonic epithelium, inducing diarrhoea. So, in a model, it should be tested whether the mice are protected against diarrhoea.
2) The authors talk about the C-terminal domain of the toxins as the “receptor binding domains”. In other papers, the C-terminal domains are usually named CROPS, standing for Combined Repetitive Oligopeptides. In any case, although the CROPs have been shown to bind to possible receptors, it has been shown that toxins lacking the mentioned domains still retain their toxicity, indicating that the receptor binding domains are not exclusive (Genesyuerek et al., Mol. Microbiol. 79, 1643-1654 (2011), Olling et al., PLoS ONE 6, e17623 (2011) and Smits et al., Nat Rev Dis Primers. 2016 Apr 7;2:16020.). Therefore, one could speculate that inducing antibodies against these domains is not sufficient to block the toxicity of TcdA and TcdB. The authors see a clear effect in their mouse model, but as said before, this model is not representative for CDI. It could well be that in the colon other receptors are used by the toxins.
3) Other papers have shown that antibodies against the CROPS (RBD) are able to inhibit toxicity of C. difficile toxins, so no real new insights are shown (Murase et al., J Biol Chem. 2014 Jan 24;289(4):2331-43.; Sauerborn et al., FEMS Microbiol Lett. 1997 Oct 1;155(1):45-54.).
Other specific remarks about the paper.
Line 55: Needs a remark about fecal microbiota transplant as very good alternative for CDI treatment.
Line 64: GTPases, plural. Several GTPase are targeted by the catalytic domain of the C. difficile toxins.
Line 65: Tha authors forget to mention the autoproteolytic domain (which has also been crystallized) that is needed for release of the catalytic domain into the cytosol.
Line 67: Description of the three dimensional structures of the toxins needs references to the papers that describe these structures (For instance papers of the Borden Lacy lab). Line 70: Several other recpetors have been recently identified (Check Smits paper)
Line 74: Reference 19 on this position is not correct.
Line 76: Strange to refer to paper about SARS coronavirus.
Line 95: positions1867-2708 needs a space after positions
Line 169: “Tissues were fixed…” What tissues? In is not described what tissue was fixed.
Judging on the figure 5, gut tissue was fixed. How can the toxin end up in the gut, when it was injected into the perineum?
Line 187: tagatthe.. needs spaces
Figure 1: A and B show no clear markers. This should be improved. In B, no band is visible.
Figure 2: According to the authors toxin A is 550 kD and Toxin B 220 kD. Toxin A size might be in the right ballpark, but Toxin B is not, it is far too small. Either the size is incorrect, or another protein was purified. Why not test the toxins on a Western blot to confirm the identity?
Line 304: “Quantitative scoring by an expert” should be elaborated. What does that mean? How was this done?
Line 321: For traditional method. What does that mean?
Line 328: It is strange to compare the toxicity of a toxin and a virus.
Line 351: strain was usually used? What does that mean?
Line 352-364 should not be part of the discussion but rather M&M and results.
Line 394-396: I do not understand this sentence.
Author Response
Dear reviewer:
We thank your constructive criticism and believe that we have addressed all of your concerns. We hope our findings have significance for those studying about receptor-binding domains of Clostridium difficile toxins can be used for vaccine development against this infection.
Response to Reviewer 2 Comments
1. You expressed your concern about “The animal model which the authors use does not mimic a proper C. difficile infection. Although the authors claim that “mice are not sensitive to CDI”, excellent mouse models have been described by for example the group of Vince Young, in which even recurrent CDI can be mimicked (see for instance Seekatz et al., Infect Immun. 2015 Oct;83(10):3838-46). The problem with the model is that purified toxins are injected into the peritoneum of the mice. In a normal CDI, the toxins are not present in the peritoneum and therefore, the model is not representative for CDI. Also, the target of the C. difficile toxins is the colonic epithelium, inducing diarrhoea. So, in a model, it should be tested whether the mice are protected against diarrhoea.”
Response1: That is really a nice question. In the “discussion” section we discussed that we do have some problem to use this standard mouse model. We use the toxin challenge model based on reference (Seregin S, et al, 2012, vaccine). we still continue the research about vaccine development of Clostridium difficile. We are trying to develop antibiotic regimes now. We will show new results in our following publications.
2. You expressed your concern about “The authors talk about the C-terminal domain of the toxins as the “receptor binding domains”. In other papers, the C-terminal domains are usually named CROPS, standing for Combined Repetitive Oligopeptides. In any case, although the CROPs have been shown to bind to possible receptors, it has been shown that toxins lacking the mentioned domains still retain their toxicity, indicating that the receptor binding domains are not exclusive (Genesyuerek et al., Mol. Microbiol. 79, 1643-1654 (2011), Olling et al., PLoS ONE 6, e17623 (2011) and Smits et al., Nat Rev Dis Primers. 2016 Apr 7;2:16020.). Therefore, one could speculate that inducing antibodies against these domains is not sufficient to block the toxicity of TcdA and TcdB. The authors see a clear effect in their mouse model, but as said before, this model is not representative for CDI. It could well be that in the colon other receptors are used by the toxins.”
Response2: Yes, we totally agree with you. We did find the clear effect in our mouse model. Tian used fusion RBD vaccine candidate to protect mice in standard animal models. We are still working for this vaccine and the details about mechanism.
3. You expressed your concern about “Other papers have shown that antibodies against the CROPS (RBD) are able to inhibit toxicity of C. difficile toxins, so no real new insights are shown (Murase et al., J Biol Chem. 2014 Jan 24;289(4):2331-43.; Sauerborn et al., FEMS Microbiol Lett. 1997 Oct 1;155(1):45-54.).”
Response3: We believe our vaccine design is optimal for developing countries. We also discussed some other vaccine designs in “discussion” section.
4. You expressed your concern about “Line 55: Needs a remark about fecal microbiota transplant as very good alternative for CDI treatment.”
Response4: Thank you for you good suggestions. We talked about “fecal microbiota transplant as very good alternative for CDI treatment” in “discussion” section and also mentioned in “introduction” section, please see line 57 and line 388.
5. You expressed your concern about “Line 64: GTPases, plural. Several GTPase are targeted by the catalytic domain of the C. difficile toxins.”
Response5: We corrected, thank you.
6. You expressed your concern about “Line 65: The authors forget to mention the autoproteolytic domain (which has also been crystallized) that is needed for release of the catalytic domain into the cytosol.”
Response6: We added it, please see line 68.
7. You expressed your concern about “Line 67: Description of the three dimensional structures of the toxins needs references to the papers that describe these structures (For instance papers of the Borden Lacy lab). Line 70: Several other receptors have been recently identified (Check Smits paper).
Response7: We cited the references in the text, please see line 82. I am so sorry, we did not find references of “Check Smits”.
8. You expressed your concern about “Line 74: Reference 19 on this position is not correct.”
Response8: We deleted this citation. Please see line 87.
9. You expressed your concern about “Line 76: Strange to refer to paper about SARS coronavirus.”
Response9: Yes, I agree with you. Most RBD vaccine researches focus on virus (SARS, MERS and so on), SARS RBD is the famous one, so I still prefer to citing this reference.
10. You expressed your concern about “Line 95: positions1867-2708 needs a space after positions”
Response10: We added spaces.
11. You expressed your concern about “Line 169: “Tissues were fixed…” What tissues? In is not described what tissue was fixed.”
Response11: It is gut, we revised our manuscript based on your suggestion, please see line 202.
12. You expressed your concern about “Judging on the figure 5, gut tissue was fixed. How can the toxin end up in the gut, when it was injected into the perineum?”
Response12: Sorry, we did not describe clearly. Toxin A and toxins B of CD are enterotoxins, they can enter gut by specific receptors through blood stream.
13. You expressed your concern about “Line 187: tagatthe.. needs spaces”
Response13: We added spaces to this section.
14. You expressed your concern about “Figure 1: A and B show no clear markers. This should be improved. In B, no band is visible”.
Response14: We improved Figure 1.
15. You expressed your concern about “Figure 2: According to the authors toxin A is 550 kD and Toxin B 220 kD. Toxin A size might be in the right ballpark, but Toxin B is not, it is far too small. Either the size is incorrect, or another protein was purified. Why not test the toxins on a Western blot to confirm the identity?
Response15: Natural toxins have big molecular weights, so we did native-PAGE, it is hard to show the molecular weight accurately. WB is a good method to confirm the identity, but the WB lines are not very clearly here. We used ELISA instead. We added some descriptions in our manuscript, please see line 255.
16. You expressed your concern about “Line 304: “Quantitative scoring by an expert” should be elaborated. What does that mean? How was this done?”
Response16: All pathology photos will be sent to a pathological company to make score by an expert first, then we will select and publish them.
17. You expressed your concern about “Line 321: For traditional method. What does that mean?”
Response17: That means the method people will think about first.
18. You expressed your concern about “Line 328: It is strange to compare the toxicity of a toxin and a virus.”
Response18: The same to question 9.
19. You expressed your concern about “Line 351: strain was usually used? What does that mean?”
Response19: That is means “normally”, we deleted the whole sentence because we moved this section to M&M based on your suggestion“20“.
20. You expressed your concern about “Line 352-364 should not be part of the discussion but rather M&M and results.”
Response20: We revised two sections, thank you for your suggestions. Please see line 116 and 314.
21. You expressed your concern about “Line 394-396: I do not understand this sentence.”
Response21: The protective data we got from the animal experiment will aid us further in the development of a safe and effective vaccine for human beings. We changed the description, please see line 488.
Reviewer 3 Report
The paper “Immunogenicity and protection from receptor-binding domains of toxins as potential vaccine candidates for Clostridium difficile” by Lou et al is a well written study examining potential C. difficile vaccine candidates.
The authors use recombinant fusion proteins based upon the receptor binding sequences of C. difficile toxins A (TcdA) and B (TcdB) as vaccines. Increasing levels of protection were observed in a mouse model when immunized with TcdB, TcdA, or both antigens.
This study would be greatly improved by showing that vaccination is beneficial in a more realistic model of C. difficile infection (CDI). There are several mouse models utilizing different mouse strains, antibiotic regimes, and C. difficile ribotypes as well as inoculum size. It seems unlikely therefore that researches need to fall back on IP injection of crude toxin preparations to test vaccine candidates.
Overall however this is a well conducted study in an area that requires continued research.
Minor points
Strain ATCC43255 is more commonly known as VPI10463.
Vaccines that target the RBD regions of C. difficile toxins have been considered for almost 30 years (see Lyerly D.M., 1990, Vaccination against lethal Clostridium difficile enterocolitis with a nontoxic recombinant peptide of toxin A.) and should be cited.
Author Response
Dear reviewer:
We thank your good comments and constructive criticism and believe that we have addressed all of your concerns. We hope our findings have significance for those studying about receptor-binding domains of Clostridium difficile toxins can be used for vaccine development against this infection.
Response to Reviewer 3 Comments
1. You expressed your concern about “This study would be greatly improved by showing that vaccination is beneficial in a more realistic model of C. difficile infection (CDI). There are several mouse models utilizing different mouse strains, antibiotic regimes, and C. difficile ribotypes as well as inoculum size. It seems unlikely therefore that researches need to fall back on IP injection of crude toxin preparations to test vaccine candidates.”
Response1: Yes, I totally agree with you about the animal models. Because we have some problem to make spores of this bacterium strain to set up standard oral challenge mouse model, we are trying to develop antibiotic regimes now. We will show new results in our following publications.
2. You expressed your concern about “Overall however this is a well conducted study in an area that requires continued research.”
Response 2:Thank you for your suggestions, we will continue the research about vaccine development of Clostridium difficile.
3. You expressed your concern about “Strain ATCC43255 is more commonly known as VPI10463.”
Response 3: This is a nice suggestion, we have made changes in our manuscript.
4. You expressed your concern about “Vaccines that target the RBD regions of C. difficile toxins have been considered for almost 30 years (see Lyerly D.M., 1990, Vaccination against lethal Clostridium difficile enterocolitis with a nontoxic recombinant peptide of toxin A.) and should be cited.”
Response 4: Thank you for your suggestion, we have cited this reference in our manuscript, please see line 62.
Round 2
Reviewer 2 Report
I have pasted my answers in the rebuttal. As can be noticed I still have serious issues with this paper. All comments are printed in blue.
Dear reviewer:
We thank your constructive criticism and believe that we have addressed all of your concerns. We hope our findings have significance for those studying about receptor-binding domains of Clostridium difficile toxins can be used for vaccine development against this infection.
Response to Reviewer 2 Comments
1. You expressed your concern about “The animal model which the authors use does not mimic a proper C. difficile infection. Although the authors claim that “mice are not sensitive to CDI”, excellent mouse models have been described by for example the group of Vince Young, in which even recurrent CDI can be mimicked (see for instance Seekatz et al., Infect Immun. 2015 Oct;83(10):3838-46). The problem with the model is that purified toxins are injected into the peritoneum of the mice. In a normal CDI, the toxins are not present in the peritoneum and therefore, the model is not representative for CDI. Also, the target of the C. difficile toxins is the colonic epithelium, inducing diarrhoea. So, in a model, it should be tested whether the mice are protected against diarrhoea.”
Response1: That is really a nice question. In the “discussion” section we discussed that we do have some problem to use this standard mouse model. We use the toxin challenge model based on reference (Seregin S, et al, 2012, vaccine). we still continue the research about vaccine development of Clostridium difficile. We are trying to develop antibiotic regimes now. We will show new results in our following publications.
Nothing done with my major concern
2. You expressed your concern about “The authors talk about the C-terminal domain of the toxins as the “receptor binding domains”. In other papers, the C-terminal domains are usually named CROPS, standing for Combined Repetitive Oligopeptides. In any case, although the CROPs have been shown to bind to possible receptors, it has been shown that toxins lacking the mentioned domains still retain their toxicity, indicating that the receptor binding domains are not exclusive (Genesyuerek et al., Mol. Microbiol. 79, 1643-1654 (2011), Olling et al., PLoS ONE 6, e17623 (2011) and Smits et al., Nat Rev Dis Primers. 2016 Apr 7;2:16020.). Therefore, one could speculate that inducing antibodies against these domains is not sufficient to block the toxicity of TcdA and TcdB. The authors see a clear effect in their mouse model, but as said before, this model is not representative for CDI. It could well be that in the colon other receptors are used by the toxins.”
Response2: Yes, we totally agree with you. We did find the clear effect in our mouse model. Tian used fusion RBD vaccine candidate to protect mice in standard animal models. We are still working for this vaccine and the details about mechanism.
Nothing done with my remark. Authors should explain the possible pitfalls of their approach. This would include discussion about the flaws of the mouse model they use (i.p. injection of toxins does not represent a natural infection) and the fact that blocking the RBD might not result in complete protection under natural infection conditions.
3. You expressed your concern about “Other papers have shown that antibodies against the CROPS (RBD) are able to inhibit toxicity of C. difficile toxins, so no real new insights are shown (Murase et al., J Biol Chem. 2014 Jan 24;289(4):2331-43.; Sauerborn et al., FEMS Microbiol Lett. 1997 Oct 1;155(1):45-54.).”
Response3: We believe our vaccine design is optimal for developing countries. We also discussed some other vaccine designs in “discussion” section.
Why is that? This is not mentioned in the text of the document. I do not see the benefit of this design for developing countries.
4. You expressed your concern about “Line 55: Needs a remark about fecal microbiota transplant as very good alternative for CDI treatment.”
Response4: Thank you for you good suggestions. We talked about “fecal microbiota transplant as very good alternative for CDI treatment” in “discussion” section and also mentioned in “introduction” section, please see line 57 and line 388.
When mentioning FMT, there should be references as well! For instance the van Nood paper in N. Eng. J. of Med. (2013) would be a good reference here
5. You expressed your concern about “Line 64: GTPases, plural. Several GTPase are targeted by the catalytic domain of the C. difficile toxins.”
Response5: We corrected, thank you.
6. You expressed your concern about “Line 65: The authors forget to mention the autoproteolytic domain (which has also been crystallized) that is needed for release of the catalytic domain into the cytosol.”
Response6: We added it, please see line 68.
An autoproteolytic
7. You expressed your concern about “Line 67: Description of the three dimensional structures of the toxins needs references to the papers that describe these structures (For instance papers of the Borden Lacy lab). Line 70: Several other receptors have been recently identified (Check Smits paper).
Response7: We cited the references in the text, please see line 82. I am so sorry, we did not find references of “Check Smits”.
The references 16 and 17 are about structures of toxins and their functions. They are not about receptors, so the references are inserted in the wrong place.
8. You expressed your concern about “Line 74: Reference 19 on this position is not correct.”
Response8: We deleted this citation. Please see line 87.
9. You expressed your concern about “Line 76: Strange to refer to paper about SARS coronavirus.”
Response9: Yes, I agree with you. Most RBD vaccine researches focus on virus (SARS, MERS and so on), SARS RBD is the famous one, so I still prefer to citing this reference.
If you must have this reference in, then you should say SARS Coronavirus, since SARS means Severe Acute Respiratory Syndrome and a syndrome does not have a receptor.
10. You expressed your concern about “Line 95: positions1867-2708 needs a space after positions”
Response10: We added spaces.
11. You expressed your concern about “Line 169: “Tissues were fixed…” What tissues? In is not described what tissue was fixed.”
Response11: It is gut, we revised our manuscript based on your suggestion, please see line 202.
12. You expressed your concern about “Judging on the figure 5, gut tissue was fixed. How can the toxin end up in the gut, when it was injected into the perineum?”
Response12: Sorry, we did not describe clearly. Toxin A and toxins B of CD are enterotoxins, they can enter gut by specific receptors through blood stream.
I have never heard of that. Do you have any references?
I rather think that the injection into the peritoneum was not done correctly and that the needle hit the gut as well, explaining the presence of the toxins in the gut.
13. You expressed your concern about “Line 187: tagatthe.. needs spaces”
Response13: We added spaces to this section.
14. You expressed your concern about “Figure 1: A and B show no clear markers. This should be improved. In B, no band is visible”.
Response14: We improved Figure 1.
Figure 1 is still messy. In 1D, the purified band (on the left) is of different size than the band on the Western blot? The markers are not aligned? Very confusing.
15. You expressed your concern about “Figure 2: According to the authors toxin A is 550 kD and Toxin B 220 kD. Toxin A size might be in the right ballpark, but Toxin B is not, it is far too small. Either the size is incorrect, or another protein was purified. Why not test the toxins on a Western blot to confirm the identity?
Response15: Natural toxins have big molecular weights, so we did native-PAGE, it is hard to show the molecular weight accurately. WB is a good method to confirm the identity, but the WB lines are not very clearly here. We used ELISA instead. We added some descriptions in our manuscript, please see line 255.
I am not convinced. The masses are strange. Why not show the ELISA data or do a W-blot? Now they show Coomassie gels? Then W-blot should be highly positive if it is the right protein.
16. You expressed your concern about “Line 304: “Quantitative scoring by an expert” should be elaborated. What does that mean? How was this done?”
Response16: All pathology photos will be sent to a pathological company to make score by an expert first, then we will select and publish them.
Unclear
17. You expressed your concern about “Line 321: For traditional method. What does that mean?”
Response17: That means the method people will think about first.
I still do not understand this sentence.
18. You expressed your concern about “Line 328: It is strange to compare the toxicity of a toxin and a virus.”
Response18: The same to question 9.
19. You expressed your concern about “Line 351: strain was usually used? What does that mean?”
Response19: That is means “normally”, we deleted the whole sentence because we moved this section to M&M based on your suggestion“20“.
20. You expressed your concern about “Line 352-364 should not be part of the discussion but rather M&M and results.”
Response20: We revised two sections, thank you for your suggestions. Please see line 116 and 314.
21. You expressed your concern about “Line 394-396: I do not understand this sentence.”
Response21: The protective data we got from the animal experiment will aid us further in the development of a safe and effective vaccine for human beings. We changed the description, please see line 488.
Remove “then”
Author Response
Response to Reviewer 2 Comments
1.You expressed your concern about “The animal model which the authors use does not mimic a proper C. difficile infection. Although the authors claim that “mice are not sensitive to CDI”, excellent mouse models have been described by for example the group of Vince Young, in which even recurrent CDI can be mimicked (see for instance Seekatz et al., Infect Immun. 2015 Oct;83(10):3838-46). The problem with the model is that purified toxins are injected into the peritoneum of the mice. In a normal CDI, the toxins are not present in the peritoneum and therefore, the model is not representative for CDI. Also, the target of the C. difficile toxins is the colonic epithelium, inducing diarrhoea. So, in a model, it should be tested whether the mice are protected against diarrhoea.”
Response1: That is really a nice question. In the “discussion” section we discussed that we do have some problem to use this standard mouse model. We use the toxin challenge model based on reference (Seregin S, et al, 2012, vaccine). we still continue the research about vaccine development of Clostridium difficile. We are trying to develop antibiotic regimes now. We will show new results in our following publications.
Nothing done with my major concern
Response 1: Sorry, this time we only focus on survival, not diarrhoea. Mice did suffered from diarrhea (we checked our records). We are still continue doing research about vaccine development of Clostridium difficile.
2.You expressed your concern about “The authors talk about the C-terminal domain of the toxins as the “receptor binding domains”. In other papers, the C-terminal domains are usually named CROPS, standing for Combined Repetitive Oligopeptides. In any case, although the CROPs have been shown to bind to possible receptors, it has been shown that toxins lacking the mentioned domains still retain their toxicity, indicating that the receptor binding domains are not exclusive (Genesyuerek et al., Mol. Microbiol. 79, 1643-1654 (2011), Olling et al., PLoS ONE 6, e17623 (2011) and Smits et al., Nat Rev Dis Primers. 2016 Apr 7;2:16020.). Therefore, one could speculate that inducing antibodies against these domains is not sufficient to block the toxicity of TcdA and TcdB. The authors see a clear effect in their mouse model, but as said before, this model is not representative for CDI. It could well be that in the colon other receptors are used by the toxins.”
Response2: Yes, we totally agree with you. We did find the clear effect in our mouse model. Tian used fusion RBD vaccine candidate to protect mice in standard animal models. We are still working for this vaccine and the details about mechanism.
Nothing done with my remark. Authors should explain the possible pitfalls of their approach. This would include discussion about the flaws of the mouse model they use (i.p. injection of toxins does not represent a natural infection) and the fact that blocking the RBD might not result in complete protection under natural infection conditions.
Response 2: I understand your worry. The i.p. injection can damage other tissues besides gut tissue. Actually, we do not know a lot about toxin structures, just followed some excellent team’s procedures. Please see two references below.
Two cell surface proteins have been implicated as receptors for TcdA. The first is sucrase-isomaltase (SI), which is a glycoprotein located in the brush border of small intestines. SI was shown to mediate the binding of TcdA to rabbit ileum (Pothoulakis et al.1996). A subsequent study performed by the same group identified glycoprotein 96 (gp96), a member of the heat shock protein family, as a binding partner for TcdA in human colonocytes (Na et al.2008). 1
Toxin A challenge experiments were performed at 14 dpi as previously described [14] by injecting 300 ng of freshly reconstituted toxin A (List Biological Laboratories Inc., Campbell, CA or Calbiochem, San Diego, CA) intraperitoneally in 100 μl (in PBS).2
1.Chandrasekaran R, Lacy DB.The role of toxins in Clostridium difficile infection. FEMS Microbiol Rev. 2017 Nov 1;41(6):723-750. doi: 10.1093/femsre/fux048Sergey S. Seregin,a Yasser A. Aldhamen,a David P.W. 2.Rastall,a Sarah Godbehere,a and Andrea Amalfitano.Adenovirus-based vaccination against Clostridium difficile toxin A allows for rapid humoral immunity and complete protection from toxin A lethal challenge in mice. 2012 Feb 14; 30(8): 1492–1501.
3.You expressed your concern about “Other papers have shown that antibodies against the CROPS (RBD) are able to inhibit toxicity of C. difficile toxins, so no real new insights are shown (Murase et al., J Biol Chem. 2014 Jan 24;289(4):2331-43.; Sauerborn et al., FEMS Microbiol Lett. 1997 Oct 1;155(1):45-54.).”
Response3: We believe our vaccine design is optimal for developing countries. We also discussed some other vaccine designs in “discussion” section.
Why is that? This is not mentioned in the text of the document. I do not see the benefit of this design for developing countries.
Resposne 3: We added some information in our manuscript, please see line 426.
4.You expressed your concern about “Line 55: Needs a remark about fecal microbiota transplant as very good alternative for CDI treatment.”
Response4: Thank you for you good suggestions. We talked about “fecal microbiota transplant as very good alternative for CDI treatment” in “discussion” section and also mentioned in “introduction” section, please see line 57 and line 388.
When mentioning FMT, there should be references as well! For instance the van Nood paper in N. Eng. J. of Med. (2013) would be a good reference here
Response 4: We added this reference, thank you, please see line 389.
5.You expressed your concern about “Line 64: GTPases, plural. Several GTPase are targeted by the catalytic domain of the C. difficile toxins.”
Response5: We corrected, thank you.
6.You expressed your concern about “Line 65: The authors forget to mention the autoproteolytic domain (which has also been crystallized) that is needed for release of the catalytic domain into the cytosol.”
Response6: We added it, please see line 68.
An autoproteolytic
Response 6: We corrected, please see line 68.
7.You expressed your concern about “Line 67: Description of the three dimensional structures of the toxins needs references to the papers that describe these structures (For instance papers of the Borden Lacy lab). Line 70: Several other receptors have been recently identified (Check Smits paper).
Response7: We cited the references in the text, please see line 82. I am so sorry, we did not find references of “Check Smits”.
The references 16 and 17 are about structures of toxins and their functions. They are not about receptors, so the references are inserted in the wrong place.
Response 7:We corrected.
8.You expressed your concern about “Line 74: Reference 19 on this position is not correct.”
Response8: We deleted this citation. Please see line 87.
9.You expressed your concern about “Line 76: Strange to refer to paper about SARS coronavirus.”
Response9: Yes, I agree with you. Most RBD vaccine researches focus on virus (SARS, MERS and so on), SARS RBD is the famous one, so I still prefer to citing this reference.
If you must have this reference in, then you should say SARS Coronavirus, since SARS means Severe Acute Respiratory Syndrome and a syndrome does not have a receptor.
Response 9:Thank you for your good suggestions, please see line 88.
You expressed your concern about “Line 95: positions1867-2708 needs a space after positions”
Response10: We added spaces.
11.You expressed your concern about “Line 169: “Tissues were fixed…” What tissues? In is not described what tissue was fixed.”
Response11: It is gut, we revised our manuscript based on your suggestion, please see line 202.
12.You expressed your concern about “Judging on the figure 5, gut tissue was fixed. How can the toxin end up in the gut, when it was injected into the perineum?”
Response12: Sorry, we did not describe clearly. Toxin A and toxins B of CD are enterotoxins, they can enter gut by specific receptors through blood stream.
I have never heard of that. Do you have any references?
I rather think that the injection into the peritoneum was not done correctly and that the needle hit the gut as well, explaining the presence of the toxins in the gut.
Response 12: The same to question 2.
13.You expressed your concern about “Line 187: tagatthe.. needs spaces”
Response13: We added spaces to this section.
14.You expressed your concern about “Figure 1: A and B show no clear markers. This should be improved. In B, no band is visible”.
Response14: We improved Figure 1.
Figure 1 is still messy. In 1D, the purified band (on the left) is of different size than the band on the Western blot? The markers are not aligned? Very confusing.
Response 14: Sorry for confusing, please see the attachment, we used different moleculer weight markers.
15.You expressed your concern about “Figure 2: According to the authors toxin A is 550 kD and Toxin B 220 kD. Toxin A size might be in the right ballpark, but Toxin B is not, it is far too small. Either the size is incorrect, or another protein was purified. Why not test the toxins on a Western blot to confirm the identity?
Response15: Natural toxins have big molecular weights, so we did native-PAGE, it is hard to show the molecular weight accurately. WB is a good method to confirm the identity, but the WB lines are not very clearly here. We used ELISA instead. We added some descriptions in our manuscript, please see line 255.
I am not convinced. The masses are strange. Why not show the ELISA data or do a W-blot? Now they show Coomassie gels? Then W-blot should be highly positive if it is the right protein.
Response 15: Please see the attachment. We did several times for toxin extract. It is hard to run a good natïve-PAGE again, so we used ELISA for identification instead. We added result of ELISA, please see line 256.
16.You expressed your concern about “Line 304: “Quantitative scoring by an expert” should be elaborated. What does that mean? How was this done?”
Response16: All pathology photos will be sent to a pathological company to make score by an expert first, then we will select and publish them.
Unclear
Response 16: We revised, please see line 374.
17.You expressed your concern about “Line 321: For traditional method. What does that mean?”
Response17: That means the method people will think about first.
I still do not understand this sentence.
Response 17: We revised this sentence, please see line 394.
18.You expressed your concern about “Line 328: It is strange to compare the toxicity of a toxin and a virus.”
Response18: The same to question 9.
19.You expressed your concern about “Line 351: strain was usually used? What does that mean?”
Response19: That is means “normally”, we deleted the whole sentence because we moved this section to M&M based on your suggestion“20“.
20. You expressed your concern about “Line 352-364 should not be part of the discussion but rather M&M and results.”
Response20: We revised two sections, thank you for your suggestions. Please see line 116 and 314.
21.You expressed your concern about “Line 394-396: I do not understand this sentence.”
Response21: The protective data we got from the animal experiment will aid us further in the development of a safe and effective vaccine for human beings. We changed the description, please see line 488.
Remove “then”
Response 21: We removed “then”, please see line 482.

Round 3
Reviewer 2 Report
See file attached

Author Response
1.I am not talking about the possible damage IP injection may cause here. I am talking about the fact that this mouse model does not represent a natural infection, so I still would like to see a paragraph in the discussion that explains the limitations of the mouse model that is used, as mentioned in the highlighted text in my previous comments.
Response 1: Sorry about to misunderstand you. We added a paragraph in the discussion about your concern. Please see line 468-472.
2.I do not see the added value of this sentence. Please remove it again. It is also not the point of my first remark (highlighted). This point was just to show that the paper is far from original.
Response 2: We removed it. Yes, I totally agree with you. Some teams developed RBD vaccines against CDI. But the vaccine designs are not the same. We discuss this, please see line 405-412.
3.This is not a satisfactory response. How can it be explained that the toxins end up in the gut, when injected in the peritoneum? The authors claim in their first rebuttal that the toxins can enter the gut through receptors and via the blood. That is the opposite direction of a natural infection. When I ask for a reference for this unknown mechanism, they do not answer. My opinion is still that during the injection, the gut of the mice was accidently hit, explaining the toxins in the gut.
Response 3: Sorry, we can not explain why the toxins can enter the gut by the opposite direction of a natural infection. We only focus on the vaccine effect in our study. We just followed other teams to use IP injection ( Sergey S. S et al, Vaccine ,2012; Scott M. B et al, Infect Immun ,2014). Sorry about this again.
4.Thank you for adding the table. Fortunately, I was able to decipher the Chinese markings of it through Google translate (see table down). I think it clarifies the toxin titers. I still think it should be added to the paper (after translating the Chinese).
Response 4: Sorry about this, we copied from our records and forgot to translate. We add some description in our manuscript. Please see line 256.
5.Unclear what this picture is. Please provide a legend. Is it a blot, a Coomassie stained gel? What is which lane?
Response 5: Sorry to confuse you. It is a native-PAGE. We used this picture to explain why we did ELISA to identify proteins instead of Western-bot. We run 5 time of native-PAGE, only one time is successful because of big molecular weights. This time which this picture showed was hard to use Western-blot to identify.